# A 1958 Isolate of Kedougou Virus (KEDV) from Ndumu, South Africa, Expands the Geographic and Temporal Range of KEDV in Africa

**DOI:** 10.3390/v13071368

**Published:** 2021-07-14

**Authors:** Petrus Jansen van Vuren, Rhys Parry, Alexander A. Khromykh, Janusz T. Paweska

**Affiliations:** 1Australian Centre for Disease Preparedness, CSIRO Health & Biosecurity, Private Bag 24, Geelong, VIC 3220, Australia; 2School of Chemistry and Molecular Biosciences, University of Queensland, Brisbane, QLD 4072, Australia; r.parry@uq.edu.au (R.P.); a.khromykh@uq.edu.au (A.A.K.); 3Australian Infectious Diseases Research Centre, Global Virus Network Centre of Excellence, Brisbane, QLD 4029, Australia; 4Centre for Emerging Zoonotic and Parasitic Diseases, National Institute for Communicable Diseases, National Health Laboratory Service, Johannesburg 2131, South Africa; januszp@nicd.ac.za; 5Centre for Viral Zoonoses, Department of Medical Virology, Faculty of Health Sciences, University of Pretoria, Pretoria 0001, South Africa; 6Faculty of Health Sciences, School of Pathology, University of Witwatersrand, Johannesburg 2050, South Africa

**Keywords:** Kedougou virus, flavivirus, arbovirus, Aedes, *Aedes circumluteolus*

## Abstract

The mosquito-borne flavivirus, Kedougou virus (KEDV), first isolated in Senegal in 1972, is genetically related to dengue, Zika (ZIKV) and Spondweni viruses (SPOV). Serological surveillance studies in Senegal and isolation of KEDV in the Central African Republic indicate occurrence of KEDV infections in humans, but to date, no disease has been reported. Here, we assembled the coding-complete genome of a 1958 isolate of KEDV from a pool of *Aedes circumluteolus* mosquitoes collected in Ndumu, KwaZulu-Natal, South Africa. The AR1071 Ndumu KEDV isolate bears 80.51% pairwise nucleotide identity and 93.34% amino acid identity with the prototype DakAar-D1470 strain and was co-isolated with SPOV through intracerebral inoculation of suckling mice and passage on VeroE6 cells. This historical isolate expands the known geographic and temporal range of this relatively unknown flavivirus, aiding future temporal phylogenetic calibration and diagnostic assay refinement.

## 1. Introduction

The *Flavivirus* genus (family *Flaviviridae*) are positive-sense RNA viruses with genome sizes between 10–11 kb classified by the International Committee on Taxonomy of Viruses (ICTV) into 53 recognized species [1]. Most flaviviruses are arthropod-borne viruses (arboviruses) transmitted to vertebrate hosts via the bite of an infected mosquito or tick vector. Mosquito-borne flaviviruses include West Nile (WNV), dengue (DENV), Zika (ZIKV), yellow fever, Japanese encephalitis viruses and tick-borne flaviviruses, including Omsk hemorrhagic fever, Kyasanur forest disease and tick-borne encephalitis viruses. Many of these flavivirus infections in humans can result in encephalitis and hemorrhagic disease and present a significant public health burden. For example, collectively, there are an estimated global 390 million cases of dengue virus infection annually, of which 96 million manifest into disease [2]. The worldwide emergence and spread of the mosquito-borne WNV and ZIKV from the historical African range into Europe, the Middle East, the Americas, West Asia and Oceania has resulted in extensive morbidity [3] and been facilitated by the expansion of mosquito vector *Aedes aegypti* [4]. Continued surveillance, both in arthropod vector species and vertebrate hosts, is required to monitor expansion in the vector-borne flavivirus geographical range and is an important tool in preparedness for the emergence and re-emergence of arboviruses.

The poorly studied Kedougou virus (KEDV) was first isolated in 1972 from a pool of 67 *Aedes minutus* mosquitoes collected from human baited traps in Kédougou, Senegal [5]. Robin et al. studied the pathogenicity of KEDV in newborn mice inoculated by the cerebral and peritoneal route and in adult mice inoculated by the same routes [5]. It was demonstrated that KEDV kills suckling mice in 5 days via cerebral inoculation and in 8 days via the peritoneal route. It is not pathogenic for weaned mice. Unlike closely related ZIKV and SPOV, no cytopathic effect was observed on the reported cells tested; African green monkey kidney (Vero) cells, rhesus monkey epithetical renal tissue (LLC-MK2), pig kidney epithelial (PS), human carcinoma epithelial (Hep2) and *A. albopictus* (unknown), nor does KEDV form plaques under carboxymethylcellulose [5]. Subsequent serological testing of Kédougou residents from 1972 and 1975 [5] indicated up to 24% of seroprevalence of KEDV.

To examine the ecology and geographic range of KEDV, we reviewed the literature surrounding the incidence of human cases (Table 1) and mosquito hosts (Table 2) of KEDV. Currently, there are only two reports examining screened human sera for KEDV indicating evidence for human exposure in a range of Senegalese towns between 1971–1990, with seropositivity generally below 10%. Additionally, one report of isolation of KEDV from a human in Bangui, the Central African Republic, is listed by the Centers for Disease Control and Prevention Arbovirus Catalog [6]. While there is evidence for KEDV infections in humans, to date, there are no reports of disease associated with KEDV.

**Table 1 viruses-13-01368-t001:** Data suggestive of human infection or exposure with KEDV between 1971–1990.

Country	Town	Date	Seroprevalence	Virus Detection	Reference
Senegal	Kédougou	1971	3/51 (6%)	-	[5]
		1975	33/138 (24%)		
Senegal	Saraya	1990	1/46 (2.17%)	-	[7]
	Silling		1/19 (5.26%)		
	Mako		3/56 (5.36%)		
	Salémata		3/73 (4.11%)		
	Khossanto		2/50 (4.00%)		
Central African Republic	Bangui	-	-	1	[6]

In addition to these reports of KEDV activity in humans, several isolations of KEDV from mosquitoes have been reported in Senegal and the Central African Republic (Table 2). Except for unidentified mosquito pools, KEDV has been isolated exclusively from *Aedes* species, including *Ae. aegypti* [8] (Table 2). The most recent report of KEDV is from *Ae. minutus* mosquitoes in Kedougou from 1996.

**Table 2 viruses-13-01368-t002:** Mosquito isolates of KEDV between 1958–1996.

Mosquito Species	Location	Date	Number of Isolates	Reference	Genbank Accession Number
*Aedes aegypti*	Kédougou, Senegal	1990	1	[8]	-
*Aedes circumluteolus*	Ndumu, South Africa	1958	1	[9,10] *	MZ218098/QWT28928
*Aedes dalzieli*	Kédougou, Senegal	1990	5	[7]	-
		1991	2		
*Aedes dalzieli*	Kédougou, Senegal	1990	5	[8]	-
*Aedes minutus*	Kédougou, Senegal	1972	1	[5]	NC_012533.1/AY632540.2
		1975	1		
*Aedes minutus*	Kédougou, Senegal	1991 to 1996	6	[11]	-
*Aedes tarsalis*	Bozo/Bangui, Central African Republic	1977	4	[12]	-
Mosquito pools	Kédougou, Senegal	1989	1	[13]	-
		1990	5		

* This study.

Initial antigenic testing indicated that KEDV cross-reacts to a low degree with mouse hyperimmune sera prepared against Usutu, WNV, Saint-Louis encephalitis, DENV (1, 2 and 3) and Powassan viruses [5]. The first genomic and phylogenetic characterization of KEDV based on 1026 nt of the NS5 gene was conducted by Kuno et al., 1998 [14]. Due to a close genetic relatedness to ZIKV, SPOV and the DENV viruses, KEDV was initially included in the DENV group based on serological relatedness, vector preference and a partial NS5 sequence [15]. However, subsequent full genome characterization of the prototype KEDV isolate DakAar-D1470 (Genbank ID: AY632540.2/NC_012533.1) [16] and further analyses of the *Flavivirus* phylogeny based on complete polyprotein sequences [17,18] indicate that KEDV belongs to its own group, named the Kedougou virus group.

We report on the isolation of KEDV from a pool of *Aedes circumluteolus* mosquitoes, collected in Ndumu, South Africa, in 1958. Ndumu is in the coastal plain of the northern KwaZulu Natal province of South Africa with hot semi-arid climate type (Köppen classification: BSh). The Ndumo game reserve (in Ndumu area) was the focus of intensive arbovirus surveillance in the 1950s and 1960s, and several viruses were detected from mosquito catches. These include the negative-sense phleboviruses Rift Valley fever virus (RVFV), Bunyamwera and Simbu viruses [19,20], the alphaviruses Ndumu [20,21], Sindbis [20] and Middelburg virus [20] and the flaviviruses SPOV, WNV and Wesselsbron viruses [9,10,20].

## 2. Materials and Methods

### 2.1. Virus Isolation and Passage History

Previously, 14 historical SPOV isolates were obtained from South African mosquitoes collected between 1958 and 1960 in the Ndumu area [9,10]. The mixed SPOV-KEDV isolate used for this study AR1071 was initially isolated following intracerebral inoculation of suckling mice for one passage. Subsequently, two passages of the mixed isolate were performed in Vero E6 (ATCC No. CRL-1586) green monkey kidney cells, from the archived lyophilized mouse brain material. The mixed nature of the culture was only discovered following unbiased deep sequencing as described below.

### 2.2. Library Preparation and Sequencing

Viral RNA was extracted from the clarified supernatants using the QIAamp^®^ Viral RNA Mini Kit (QIAGEN, Hilden, Germany), followed by cDNA preparation as described previously [22]. Sequencing libraries were prepared using the Nextera DNA library preparation kit as recommended by the manufacturer (Illumina, San Diego, CA, USA) and sequenced using the MiSeq Illumina platform. Raw data are available in the Sequence Read Archive (SRA) under Bio-Project accession number PRJNA501801.

### 2.3. Virus Assembly, Annotation and Phylogenetic Analyses

Random hexamer and adapter sequences were removed from the basecalled fastq files using Cutadapt v1.21, resulting in 1,273,559 clean reads [23]. Clean reads were assembled using SPAdes [24] (v3.12.0, careful flag). The SPAdes assembled contigs were queried against a representative Flavivirus database using BLASTn. The polyprotein of the Ndumu-AR1071 KEDV isolate was predicted using the Open Reading Frame Finder (https://www.ncbi.nlm.nih.gov/orffinder/; Accessed 26 April 2021). For phylogenetic placement within the *Flavivirus* genus, 40 representative polyprotein sequences were downloaded from Genbank and aligned using MAFFT (v7.475, E-INS-i method). The resultant multiple sequence alignment (40 × 3852 aa) was used to construct a consensus maximum-likelihood phylogenetic tree using IQ-TREE2 (v2.1.2) using the LG+F+R6 protein substitution model as selected using the Bayesian Information Criterion in the IQ-TREE2 ModelFinder [25]. The Ndumu-AR1071 KEDV genome sequence has been deposited in the NCBI Genbank (accession number MZ218098).

## 3. Results and Discussion

All flavivirus genomes are between ~10–11 kb and usually encode a single open reading frame that is flanked by 5′ and 3′ untranslated (UTR) regions of ~100 and ~400–700 nt, respectively. The SPAdes assembled KEDV genome is 10,555 nt, and for manual validation of the KEDV genome, 1703 (0.13%) reads (1554 paired) were remapped to the reference using BWA-MEM (v0.7.13-r1126) [26] with default settings. This resulted in an average coverage of 34.82× for the whole assembly and a minimum/maximum read depth of 4/168 for the coding region (Figure 1A). The resultant SAM file was sorted and converted to a BAM file containing only mapped reads using bedtools (v1.3) [27] and inspected using the Integrative Genomics Viewer (v2.3) [28].

The Ndumu-AR1071 KEDV genome sequence has a 5′UTR of 107nt, which is almost entirely identical (99.05%) to the KEDV DakAar-D1470 strain (Genbank: NC_012533.1). We could only recover 221 nt of the 3′UTR (compared to 390 nt of DakAar-D1470), which was quite divergent to the DakAar-D1470 strain with 87.03% pairwise nucleotide identity. Overall, the complete sequence bears 80.51% pairwise nucleotide identity to the prototype DakAar-D1470 strain 10,723 nt. The 3′UTR of flaviviruses contain highly structured, nuclease resistant elements [29]. However, given the incompleteness of the 3′UTR, we did not attempt to draw the structured elements of the 3′UTR. We were also unable to extend the 3′UTR through mapping to the DakAar-D1470 strain, suggesting divergent 3′UTR elements.

The predicted polyprotein of the Ndumu-AR1071 KEDV genome has a length of 3408 aa which is identical to the size previously reported (Genbank: YP_002790882.1) [16]. Pairwise alignment between both KEDV strains indicated a 93.34% pairwise amino acid identity with 139 differences. However, there were no changes in identified polyprotein post-translational cleavage locations previously reported for KEDV, which is important for virion assembly, secretion and infectivity.

Kedougou virus was initially classified as a member of the dengue virus subgroup [15] before the characterization of its complete genome [16], following which it was classified in its own group [1]. Given the lack of sequencing data of KEDV strains, temporal phylogenetic inferences dating the KEDV clade using the time to most recent common ancestor (TMRCA) estimates would be misleading. Instead, we constructed phylogenetic inferences using the predicted polyprotein of the SA isolate of KEDV against the *Flavivirus* genus. Phylogenetically, we were able to reconstruct the overall topologies of this genus; as previously reported [17,18], KEDV forms a close but separate clade, sitting between the distinct clades representing the Spondweni serogroup (SPOV and ZIKV) and the dengue virus group (Figure 1B).

Despite being isolated from *Aedes* spp. mosquitoes on several occasions between 1958 and 1991 (Table 1), Ndumu-AR1071 represents only the second isolate of Kedougou virus with a published complete or coding complete sequence. Considering the genetic divergence between isolates of this virus sampled 14 years apart (1958/1972) from two extreme geographical locations on the African continent (Kedougou, Senegal in western Africa, and Ndumu in South Africa), there is a major gap in our knowledge of the true genetic diversity and evolution of KEDV and how this relates to temporal and geographical separation. The level of genetic divergence between the two KEDV isolates is surprising given that isolates of SPONV that were characterized in the same study [9] are 50 years apart and only contained a divergence of pairwise nucleotide identity of 0.31% to 2.25% to the 2016 SPONV Haiti isolates. Further studies characterizing the genetic identity from central African isolates and phenotypic characterization of pathogenicity are required.

The importance of KEDV to human health is not well understood. Serological surveillance studies performed between 1971 and 1990 in Senegal and the Central African Republic around mosquito capture sites suggest that humans were exposed to KEDV, presumably through infected mosquitoes, with seropositivity rates of up to 24% noted (Table 2). This is not surprising considering that the prototype KEDV was isolated in 1972 from *Ae. minutus* caught by human baited mosquito traps [5].

The mosquito host of the South African KEDV isolate is the *Ae. circumluteolus* mosquito, which breeds in temporary flood ponds [30] and was shown to reach exceedingly high levels seven or more days after the Usutu or Pongola rivers inundated the flood plains in the Ndumu game reserve area [30]. Within this region, *Ae. circumluteolus* is the most commonly caught mosquito species [20]. It is thought that mosquito numbers are related mainly to river flooding; however, there is also a weaker association with local rainfall [30]. The *Ae. circumluteolus* mosquito is a competent vector for many arboviruses, including the bunyaviruses Bunyamwera virus [19], RVFV [31] and Pongola virus [32]. Surveillance and experimental infection studies have also demonstrated *Ae. circumluteolus* to be a competent vector of the flaviviruses Wesselsbron virus [20,33,34] and SPONV [9,10,20] and also the dsRNA Orbivirus, Lebombo virus [35].

To understand the ecology and transmission cycles of KEDV in Africa, further information is required about the mosquito range, feeding preference and habitat of mosquito vectors and suitable vertebrate hosts. The *Ae. circumluteolus* mosquito is a sylvatic mosquito. While experimentally demonstrated to take blood meals from human hosts, the preferred host is bovines such as antelope or cattle [30,36,37] and to a lesser extent, horses and zebra [37]. In blood meal screening studies conducted on the mosquito identified with KEDV in the Central African Republic [12], *Ae. tarsalis* indicated that the mosquito takes blood meals preferentially from bovines such as cattle [38], the bushbuck antelope [39] and also, to a lesser extent, other livestock such as sheep and goats [38]. In comparison, not much is known of the host preferences of the West African mosquito vectors of KEDV, *Ae. dalzieli* and *Ae. minutus*; both do take blood meals from human hosts; and finally, the domestic *Ae. aegypti* is well reported to preferentially prefer blood meals from humans over any other mammals [40]. In contrast, most of the mosquitoes identified as positive for KEDV are floodwater mosquitoes; both *Ae. dalzieli* and *Ae. aegypti* are considered tree-hole mosquitoes with larvae found in water-holding containers and discarded containers such as tires [41]. Given the collective information of these mosquitoes’ known host preferences and domestic status, we postulate that KEDV is likely to exist almost exclusively sylvatically through a bovine–mosquito transmission cycle. While most KEDV is likely horizontally transmitted between mosquito and vector host, given that *Aedes* eggs are desiccant resistant and can remain in a dormant state for months or years, hatching following sustained submersion; the transovarial transmission of viruses between successive generations may also be possible (reviewed in [42]).

## 4. Conclusions

Retrospective genomic characterization of historical arbovirus isolates at the National Institute for Communicable Diseases arbovirus repository has led to discovering a mixed virus isolate containing AR1071-SPOV and AR1071-KEDV. This represents the earliest known isolation of this virus and expands the known geographical range of this poorly studied virus.

Although there is no record of human disease resulting from KEDV infection to date, serological data suggest that humans have been exposed in areas where the virus activity was reported. The close genetic relatedness of Ndumu-AR1071 KEDV to flaviviruses with severe impact on public health and evidence of the potential role of *Ae. aegypti* in the transmission of KEDV [8] should not be ignored. The worldwide expansion of *Ae. aegypti* [4] has facilitated the expansion of ZIKV and DENV in previously naïve areas of the world and placed a significant burden on public health systems. In this context, monitoring of KEDV activity and evolution in the African human population must be considered. Moreover, co-occurrence of KEDV with other flaviviruses could complicate diagnostic and serological surveillance for important flaviviruses in patients with suspected arboviral infection. Consequently, African investigators involved in arbovirus surveillance and diagnosis should consider KEDV to better understand its distribution and potential public health importance.

## Figures and Tables

**Figure 1 viruses-13-01368-f001:**
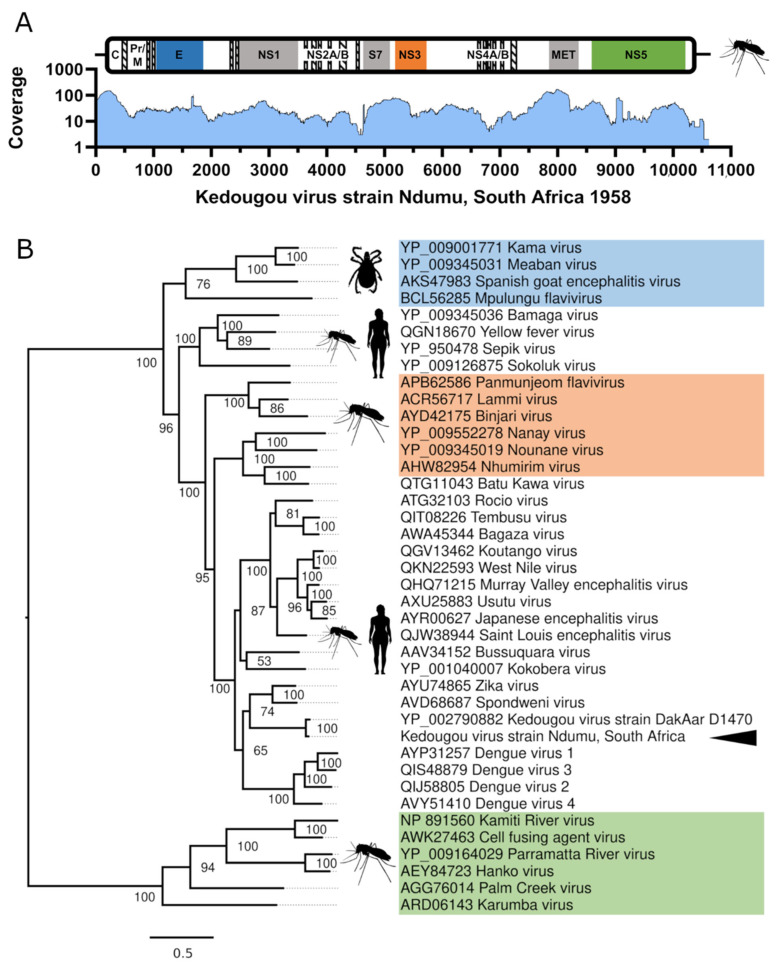
Genome coverage and phylogenetic position of the Kedougou virus isolate (Ndumu, South Africa). (**A**) Genome schematic showing protein domains as colored boxes, C; capsid, prM; pre-membrane, E; Flavivirus envelope glycoprotein, NS; non-structural proteins, pr; flavivirus propeptide, M; flavivirus envelope glycoprotein M; S7; flavivirus NS3 serine peptidase S7, MET; FtsJ-like methyltransferase. Coverage was obtained using samtools depth and plotted using Graphpad Prism (v9.0). (**B**) Consensus maximum-likelihood tree of representatives of the *Flavivirus* genus constructed using IQ-TREE v2.1.2 and the LG+F+R6 amino acid substitution model with 1000 bootstraps. The insect only classical insect flavivirus clade is indicated in green, tick flavivirus clade is indicated in blue and lineage two insect only clade shown in orange. The AR1071 Ndumu Kedougou virus isolate is marked with the arrowhead.

## Data Availability

The final sequence of KEDV-SA has been deposited in GenBank under accession number MZ218098. Raw high-throughput sequencing data are available in the Sequence Read Archive (SRA) under the accession SRR8133411.

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
