# Peer review of "A 1958 Isolate of Kedougou Virus (KEDV) from Ndumu, South Africa, Expands the Geographic and Temporal Range of KEDV in Africa"

_viruses, 2021, doi:10.3390/v13071368_

Round 1
Reviewer 1 Report
This paper describes the complete genomic sequence of a 1958 isolate of Kedougou virus. This isolate was from Ndumu, KwaZulu-Natal, in South Africa, and was the second full-length sequence of Kedougou virus to be reported, the other being from Senegal. The paper extends the known geographic range of the virus. The close relationship of Kedougou to other important human pathogens, especially the dengue viruses, Spondweni and Zika viruses, provides a strong reason for a greater understanding of the geographic range and pathogenesis of this virus, and thus this paper assists with the former, and is an important addition to the flavivirus literature.
There is one important typographical change to be made; Table 1. Mosquito isolates of KEDV between 1958-1996 needs to be changed to Table 2. Mosquito isolates of KEDV between 1958-1996.
Author Response
We thank reviewer 1 for spotting this mistake, and it has been corrected.
Reviewer 2 Report
Overview
In this short communication, authors describe the coding-complete genome sequence of a 1958 isolate of KEDV from a pool of Aedes circumluteolus mosquitoes collected in South Africa, which expands what was previously known about the geographic and temporal range of this relatively obscure and heretofore poorly-characterized flavivirus. The value in this study lies not only in laying a groundwork for examining the evolution of this virus over time, but in improving our understanding of the ecology and pathogenesis (or lack thereof) of flaviviruses co-circulating with flaviviruses that are known to cause morbidity and mortality in human and animal populations. Owing to difficulties in flavivirus diagnostics, improved data on the genetics and ecology of less well-characterized flaviviruses will also aid in improvement of flavivirus diagnostics.
Initial grouping of KEDV with the DENV viruses prior to a more thorough sequencing and phylogenetic analyses of genus Flavivirus emphasizes the importance of both classical virological techniques and whole-genome sequencing and phylogenetic analyses in the discovery and characterization of new viruses. Interestingly, seroprevalence studies carried out at the time of initial isolation suggest regular exposure of humans to this virus. To synthesize what is known about KEDV, authors also reviewed the literature surrounding human cases (or human exposure) and mosquito hosts.
Specific items:
Lines 56-59: No CPE is appreciated in most cell lines that sometimes form plaques upon infection with other closely-related flaviviruses (e.g. ZIKV in Vero cells) In the discussion, authors could comment on how CPE, or lack thereof, of KEDV on these different cell lines compares to the in vitro growth kinetics of Zika, Spondweni, and other closely-related flaviviruses.
Line 63: Should read “evidence for human exposure” since this table also includes seroprevalence studies in absence of disease. Individuals can seroconvert without mounting a productive infection featuring replicating virus.
Lines 99-105: Could you elaborate on how the KEDV isolate described was isolated from the mixed SPOV-KEDV isolate that was initially isolated? Was it plaque-purified prior to further passage on Vero cells prior to sequencing?
Lines 139-146: It’s interesting that the 3’ UTR was so divergent between the DakAar-D1470 strain and the strain characterized in this publication. How does this compare to divergence of different strains of ZIKV, or of Spondweni? Given what is known about the 3’ UTR playing a role in the evasion of innate immunity, it is of interest to consider what evolutionary advantages SNPs in the 3’ UTR may have over point mutations in the coding sequence.
Lines 185-187: Differentiating serologic status from true infection is important. Rather than serology demonstrating that the virus was transmitted to humans, it is more appropriate to say in line 186 that serologic studies indicate that humans were exposed.
Line 189: This is a very interesting and relevant point, and it would be interesting to discuss what is known about bloodmeal analysis and host bloodmeal preference of the Aedes spp. from which this virus has been isolated in the past. This could guide future animal surveillance studies. Is this predominantly a sylvatic cycle? What other pieces of information must be investigated to better characterize the ecology of this heretofore poorly-described virus?
Lines 200-203: Echoing previous comment; this is fascinating from an ecological perspective, and it would be of interest to expand upon these vertebrate-vector associations for the other mosquito species KEDV has been isolated from. Interesting that the virus has only been isolated from Aedes species, and in line with what we know about other flaviviruses in this geographic region and this phylogenetic clade. What do we know about these specific Aedes species related to ecological niche, host preference, and bloodmeal data?
Line 210: Change from “humans have been infected” to “humans have been exposed”
Table 1: Change heading to “Data suggestive of human infection or exposure with KEDV between 1971-1990”. Add column next to seroprevalence for “Viral detection” or “Virus isolation” (whichever you’d prefer to use to describe the case in Bangui).
Very nicely-written paper.
Author Response
Lines 56-59: No CPE is appreciated in most cell lines that sometimes form plaques upon infection with other closely-related flaviviruses (e.g. ZIKV in Vero cells) In the discussion, authors could comment on how CPE, or lack thereof, of KEDV on these different cell lines compares to the in vitro growth kinetics of Zika, Spondweni, and other closely-related flaviviruses.
Response: Considering that our KEDV isolate is a mixed KEDV/SPOV isolate, and SPOV causes CPE in VeroE6 cells, we are unable to comment on whether our isolate causes CPE. The statement in lines 56 – 59 refers to published data. Because we do not discuss our KEDV isolate replication kinetics at all in the discussion (this was not studied), and there is no elegant way to insert a comment about CPE compared to other flaviviruses in the current version of the discussion, we have inserted a small comment into the introduction where the published pathogenicity is discussed, to mention the difference with ZIKV and SPOV.
Line 63: Should read “evidence for human exposure” since this table also includes seroprevalence studies in absence of disease. Individuals can seroconvert without mounting a productive infection featuring replicating virus.
Response: The sentence has been amended as suggested.
Lines 99-105: Could you elaborate on how the KEDV isolate described was isolated from the mixed SPOV-KEDV isolate that was initially isolated? Was it plaque-purified prior to further passage on Vero cells prior to sequencing?
Response: We did not attempt to plaque purify the mixed isolate to obtain purified isolates. The discovery that the culture contains KEDV, in addition to SPOV, was purely incidental because of analysis of unbiased deep sequencing analysis. Identification and assembly of the KEDV sequence was performed as described in the manuscript. A sentence has been added to material and methods to make this clearer.
Lines 139-146: It’s interesting that the 3’ UTR was so divergent between the DakAar-D1470 strain and the strain characterized in this publication. How does this compare to divergence of different strains of ZIKV, or of Spondweni? Given what is known about the 3’ UTR playing a role in the evasion of innate immunity, it is of interest to consider what evolutionary advantages SNPs in the 3’ UTR may have over point mutations in the coding sequence.
Response: We appreciate how significant the 3’UTR is for Flavivirus pathogenesis. Given the nucleotide divergence of the 3’UTR identified by the reviewer we re-analysed the 3’UTR. In doing so we identified a 3’UTR duplication in the sequence, suggesting misassembly. This new 221nt 3’UTR is now more similar to the Dakar strain (87.03%). While it would be nice to draw the structural elements of the 3’UTR given the length and low coverage of mapped reads we would not be confident in structures predicted for the SA strain and therefore would not be comfortable comparing between the Dakar and SA strains. We appreciate the reviewer’s comments here as we would not have picked that up otherwise. All locations where the genome size is mentioned have now been changed in text.
Lines 185-187: Differentiating serologic status from true infection is important. Rather than serology demonstrating that the virus was transmitted to humans, it is more appropriate to say in line 186 that serologic studies indicate that humans were exposed.
Response: This section of the text has been amended as per the reviewer’s recommendation.
Line 189: This is a very interesting and relevant point, and it would be interesting to discuss what is known about bloodmeal analysis and host bloodmeal preference of the Aedes spp. from which this virus has been isolated in the past. This could guide future animal surveillance studies. Is this predominantly a sylvatic cycle? What other pieces of information must be investigated to better characterize the ecology of this heretofore poorly described virus?
Response: We postulate that it’s likely that Kedougou virus is maintained in a sylvatic cycle but also possible there is some level of transovarial transmission. We have added some further details about the mosquito species that may inform the ecology and potential transmission of the virus as well as the host blood meal status (Lines 203-224) as a reasonable primer of KEDV transmission in Africa.
Lines 200-203: Echoing previous comment; this is fascinating from an ecological perspective, and it would be of interest to expand upon these vertebrate-vector associations for the other mosquito species KEDV has been isolated from. Interesting that the virus has only been isolated from Aedes species, and in line with what we know about other flaviviruses in this geographic region and this phylogenetic clade. What do we know about these specific Aedes species related to ecological niche, host preference, and bloodmeal data?
Response: As per above, we’ve integrated details about the ecology and blood meal preference status where we could from the literature (lines 203-224). Unfortunately, this is a little patchy in the literature, at least to our knowledge so we would not like to extrapolate this information.
Line 210: Change from “humans have been infected” to “humans have been exposed”
Response: Amended as suggested.
Table 1: Change heading to “Data suggestive of human infection or exposure with KEDV between 1971-1990”. Add column next to seroprevalence for “Viral detection” or “Virus isolation” (whichever you’d prefer to use to describe the case in Bangui).
Response: Amendments made as suggested